# Prevalence and Mechanisms of Skeletal Muscle Atrophy in Metabolic Conditions

**DOI:** 10.3390/ijms24032973

**Published:** 2023-02-03

**Authors:** Lauren Jun, Megan Robinson, Thangiah Geetha, Tom L. Broderick, Jeganathan Ramesh Babu

**Affiliations:** 1Department of Nutritional Sciences, Auburn University, Auburn, AL 36849, USA; 2Boshell Metabolic Diseases and Diabetes Program, Auburn University, Auburn, AL 36849, USA; 3Department of Physiology, Laboratory of Diabetes and Exercise Metabolism, College of Graduate Studies, Midwestern University, Glendale, AZ 85308, USA

**Keywords:** skeletal muscle, muscle atrophy, obesity, sarcopenia, diabetes, Alzheimer’s disease

## Abstract

Skeletal muscle atrophy is prevalent in a myriad of pathological conditions, such as diabetes, denervation, long-term immobility, malnutrition, sarcopenia, obesity, Alzheimer’s disease, and cachexia. This is a critically important topic that has significance in the health of the current society, particularly older adults. The most damaging effect of muscle atrophy is the decreased quality of life from functional disability, increased risk of fractures, decreased basal metabolic rate, and reduced bone mineral density. Most skeletal muscle in humans contains slow oxidative, fast oxidative, and fast glycolytic muscle fiber types. Depending on the pathological condition, either oxidative or glycolytic muscle type may be affected to a greater extent. This review article discusses the prevalence of skeletal muscle atrophy and several mechanisms, with an emphasis on high-fat, high-sugar diet patterns, obesity, and diabetes, but including other conditions such as sarcopenia, Alzheimer’s disease, cancer cachexia, and heart failure.

## 1. Introduction

Skeletal muscle is the largest organ in the human body and accounts for more than 40% of body mass. This essential tissue provides physical structure, mobility, and protection to vital organs in the body, and it aids in regulating body temperature and basal metabolic rate. Skeletal muscle is comprised of distinct muscle fiber subtypes categorized by the presence of myosin heavy chain (MyHC) structures and physiologic capabilities [1]. Human skeletal muscle fibers are generally described as either slow-twitch (mostly oxidative) or fast-twitch (mostly glycolytic), meaning they have different sensitivities and responses to specific stimuli and protein turnover rates [2]. For example, it is accepted that degradation of glycolytic muscle fibers (type IIx and type IIb) is linked to diabetes and aging. In contrast, oxidative muscle fibers (type I) are more sensitive to inactivity and denervation-induced atrophy [2].

Muscle atrophy is defined as a loss of the tissue [3]. Atrophy occurs when the rate of protein degradation exceeds the rate of synthesis. This loss is highly associated with high-fat, high-sugar diet patterns, generally known as the Western-style diet, and chronic conditions such as obesity, diabetes, Alzheimer’s disease, aging, cachexia, and heart failure. Muscle atrophy during these chronic conditions should not be neglected because a decrease in muscle mass has been associated with a poor quality of life from the increased risk of disability, decreased functional capacity, basal metabolic rate, and bone mineral density [4]. Consequently, muscle loss during aging, termed sarcopenia, has been associated with increased morbidity and mortality rates [5]. Therefore, understanding the prevalence and mechanisms of muscle atrophy in different physiological conditions has become a topic of interest. Herein, the fundamental mechanism of skeletal muscle atrophy will be discussed. Additionally, this review will discuss the mechanisms of muscle atrophy under the Western-style diet pattern and in various metabolic conditions.

## 2. Pathophysiology of Skeletal Muscle Atrophy

### 2.1. High-Fat Diet

The chronic overconsumption of Western-style diets has long been a leading cause of obesity and other comorbidities, including cardiovascular and cardiometabolic diseases, hypertension, metabolic syndrome, T2DM, and kidney disease [6]. This Western-style diet pattern is a supersized portion containing a high percentage of fat and refined sugar. This diet, coupled with a sedentary lifestyle, has become a significant contributing factor to the national obesity epidemic [7]. In particular, high saturated fat intake has been known to result in fat mass gain and elevated comorbidity factors such as glucose intolerance, hyperlipidemia, [8], and insulin resistance [9]. Additionally, studies have linked high-fat diets (HFD) and high-sugar diets (HSD) with decreased skeletal muscle mass and functionality [10] in a fiber-specific and sex-dependent manner [11,12]. 

In mouse studies, HFD was associated with a significant body composition change. Increased body mass index (BMI) and lipid accumulation in the liver and skeletal muscle [12,13,14] are observed in mice fed a HFD. These morphological changes are observed in all muscle types more quickly in aged mice than in their young counterparts, with glycolytic type II muscle fibers being the least responsive [12]. Similarly, in a middle-aged mouse model, HFD was associated with a significant loss of grip strength and sensorimotor coordination. This was associated with reduced myofiber area, satellite cells, and myonuclei of the glycolytic gastrocnemius muscles [15]. Morphological changes in this muscle were consistent with an atrophic phenotype related to the diet, suggesting that HFD is an independent risk factor for muscle atrophy and may worsen sarcopenia-associated muscle atrophy.

The previous studies indicate that muscle atrophy occurs in a fiber-type-specific and sex-dependent manner when exposed to a short-term or long-term HFD diet. Indeed, short-term (3–4 weeks) HFD showed muscle atrophy in the oxidative slow-twitch (type I) muscles such as soleus [16,17], but not in the glycolytic fast-twitch (type II) extensor digitorum longus (EDL) and gastrocnemius [15] muscles. Additionally, oxidative soleus muscles produced significantly less force and had larger intramyocellular lipid droplets, consistent with increased lipid metabolism [18]. Furthermore, the soleus muscles of the short-term HFD mice showed upregulation of mRNA levels for inflammatory cytokines (IL-6, TNF-α), ubiquitin ligases (MuRF1, Atrogin-1), and markers of oxidative stress [10,19]. Correspondingly, loss of mitochondria and mitochondrial enzyme activity was also observed in the soleus muscles of short-term (3 weeks) HFD-fed mice [17]. These results indicate that HFD enhances the activation of protein degradation primarily in the oxidative muscle fibers first. Interestingly, this loss of soleus muscle mass and strength was predominant until 8 weeks of HFD administration [20].

A long-term (12 weeks or more) HFD results in an even greater rate of muscle depletion [14,16,17], especially in the glycolytic muscles, such as tibialis anterior [21] and EDL muscles, as compared to generally oxidative soleus muscles [20]. In addition, a long-term HFD decreased contractile force associated with changes in the percentage of type II fibers, mitochondrial oxidative enzyme activity, and intramyocellular lipid levels [16], suggesting a shift in the muscle fiber type during the catabolic state. This finding is in agreement with other investigations that observed a tendency to shift from glycolytic toward oxidative muscle fibers during long-term HFD feeding [16], which corresponds well with the muscle fiber-type shift in response to nutrient- or inflammatory-related conditions [21]. This report suggests that a change in the ratio of fiber types may be one of the mechanisms resulting in the decreased muscle mass and strength associated with a long-term HFD.

The increased percentage of oxidative muscles resulting from a long-term HFD can be partially explained by nutrient utilization and oxidative capacity of the oxidative muscle fibers. The dynamic increase in atrophic markers in the starting phase of the HFD indicates a perturbation in which the oxidative muscles must adapt in order to prevent damage and conserve integrity [10]. This is in line with HFD feeding resulting in a progressive increase in mitochondrial content and higher capacity to oxidize lipid substrates [13]. Under normal conditions, oxidative muscles rely mainly on lipid metabolism, whereas glycolytic muscles are highly dependent on glucose metabolism [11]. Furthermore, oxidative muscles are more resistant to fatigue and obtain energy through oxidative metabolism [11]. According to previous reports, oxidative muscles are affected during the initial phase of HFD feeding. Still, oxidative muscle seems to adapt to the greater lipid accumulation by decreasing the lipogenic enzymes and increasing the lipolytic enzymes. Thus, the oxidative tissues seem unaffected after the adaptation phase while the glycolytic muscles are depleted [11].

In contrast to the previous investigations, Campbell and colleagues reported that 16 weeks of HFD did not affect skeletal tissue morphology of female rats [22]. This discrepancy could be explained by several factors in the experimental model, namely by age, sex, and physical activity level. Another study revealed that HFD-fed female mice show better antioxidant protection and generally maintain a better insulin sensitivity profile [11]. These effects may be partly due to the types of sex hormones present as well as differences in body fat distribution and adipokine levels, which regulate mitochondrial biogenesis of skeletal muscle [11]. Furthermore, in female rats, higher fat storage by adipose tissue together with a higher fatty acid oxidation capacity in the muscle led to more efficient handling of energy excess when compared with male rats [23]. In a similar way, the level of physical activity may determine the rate of HFD diet-induced muscle atrophy. Indeed, long-term HFD mice with normal cage activity maintained their muscle mass, whereas HFD mice with sedentary cage activity showed a decrease in muscle mass [14].

Several mechanisms of HFD-mediated skeletal muscle atrophy have been discussed. Insulin-like growth factor-1 (IGF-1) is a major growth factor that regulates both anabolic and catabolic processes in skeletal muscle. During the anabolic process, IGF-1 activates the IGF-1/serine-threonine protein kinase (Akt)/mammalian target of rapamycin (mTOR) pathway, resulting in protein synthesis. Simultaneously, IGF-1/Akt can inhibit translocation of the Forkhead box O (FoxO) class of transcription factors which inhibits the transcription of E3 ubiquitin ligases, thereby preventing protein breakdown. Generally, HFD causes lipid accumulation in the skeletal muscle, mitochondrial dysfunction, insulin resistance, and decreases the proteins necessary for the IGF-1/Akt/mTOR signaling pathway [24]. Thus, HFD is an independent factor leading to disruption in this proteogenic pathway [18]. In fact, this was associated even with creatinine supplementation, which is a known upregulator of the IGF-1 signaling pathway [24]. In this study, creatine supplementation upregulated proteins involved in the IGF-1-Akt signaling pathway; however, these proteins were downregulated upon HFD feeding [24]. This finding suggests that dietary fat content is crucial to muscular hypertrophy and its function.

A HFD was also shown to elevate the expressions of atrophy-related proteins such as muscle ring finger (MuRF)-1 and muscle atrophy F-box (MAFbx) in skeletal muscle via the FoxO transcription factor or activation of the nuclear factor kappa B (NFκB) pathway [25]. NFκB is a critical regulator of immune and inflammatory responses which induces the expression of many proinflammatory cytokines (tumor necrosis factor (TNF)-α, interleukin (IL)-1β, IL-6, IL-8, and IL-12). This pathway has been shown to be associated with a HFD in animal studies. For instance, Carlsen et al. used NFκB luciferase reporter mice and showed that NFκB activity was higher in the HFD mice [26]. In accordance, the phosphorylation of NFκB was significantly increased in the muscles of HFD-fed mice [25], supporting the inflammation activation and the inflammation-induced dysfunction and atrophy in skeletal muscle.

Similarly, a long-term HFD was shown to increase levels of circulating glucocorticoids (GC) [27], a class of steroid hormones that participate in numerous physiological processes, including nutrient metabolism, through the classic glucocorticoid receptor (GR) [28]. In humans, cortisol is the dominant GC secreted by the adrenal cortex, and excessive levels of cortisol have been implicated in many chronic conditions such as osteoporosis, hypertension, insulin resistance, and diabetes mellitus [29]. Cortisol induces muscle atrophy by stimulating proteolysis of contractile proteins and the mobilization of amino acids, particularly in the fast-twitch type II muscle fibers [30]. Thus, an increase of blood cortisol also explains the reduced ratio of glycolytic muscle fibers after a long-term HFD consumption. 

Importantly, a Western-style diet consisting of high-fat, high-sugar foods is known to contribute to chronic disease progression. According to many cross-sectional studies, the Western diet pattern was shown to increase the risk of obesity [31], insulin resistance and diabetes [32], sarcopenia [33], Alzheimer’s disease [34], tumor growth and cancer cachexia [35], and cardiovascular diseases [36]. There is a growing body of evidence that provides better insight into the effect of the Western-style dietary pattern in the development of chronic illnesses that are associated with skeletal muscle atrophy. Therefore, more attention is needed in diet intervention as a muscle atrophy prevention strategy. The prevalence of muscle atrophy in the aforementioned chronic illnesses is described in the following sections.

### 2.2. Obesity

Obesity, defined as abnormal or excessive fat accumulation or a BMI of more than 30 kg/m^2^, is an increasingly prevalent metabolic disease in Western societies, affecting both adults and children. Rates of obesity have increased at an alarming rate in the US over the past three decades [37]. Approximately 66% of American adults are overweight or obese, conditions which are more prevalent in minority and low-socioeconomic-status groups [37]. Globally, obesity has become an important clinical and public health burden. By 2030, the number of overweight and obese adults is projected to be 1.35 billion and 573 million, respectively [38]. Economically, the cost of management of obesity and its complications has been estimated at roughly USD 2 trillion annually or 2.8% of global GDP [39]. The prevalence of obesity has been attributed to the Western diet and a sedentary lifestyle [37]. With the large availability of highly processed energy-dense foods, people are becoming obese at a younger age [40]. 

Obesity is an independent risk factor for the development of chronic diseases such as diabetes [41], cancer [42], cardiovascular diseases [43], as well as neurological, respiratory, gastrointestinal, reproductive, and psychosocial complications [39]. Thus, obesity management is a vital topic for discussion to prevent development of these life-threatening diseases. Interestingly, the risks of these chronic conditions are compounded by corresponding high levels of fat mass and low levels of muscle mass and or strength. 

The increase in fat mass partly comes from the deposition of fat between the muscle fibers, also known as intramuscular fat. Intramuscular fat is implicated in obese adults [44] and it is a particular topic of interest as it plays a significant role in skeletal muscles. For instance, intramuscular fat can negatively affect insulin-mediated glucose uptake as well as fat peroxidation [45]. Thus, this can result in insulin resistance in obese individuals [46]. As such, genetically obese mice exhibit marked hyperglycemia [47] and hyperinsulinemia [48]. Importantly, disruption in glucose utilization caused by decreased insulin sensitivity and response [49] is known to further aggravate skeletal muscle atrophy in obese humans and animals [50]. Interestingly, one study reported that diet-induced obesity alone does not induce muscle wasting or contractile dysfunction in a mouse model [14]. In fact, muscle atrophy, associated with obesity and insulin resistance, is coupled with other comorbidities such as age, heart disease, renal failure, and neuropathy [14]. In the same way, a sedentary lifestyle, which is highly prevalent in obese individuals [51], is strongly associated with significant muscle atrophy [52].

Intriguingly, increased adiposity contributes to greater muscle mass in obese individuals. As such, obese adolescents were demonstrated to have a greater absolute maximum muscle strength than non-obese counterparts mainly due to the adipose tissues acting as chronic overload stimulus on the antigravity muscles in the lower extremities [53]. When maximum muscular strength is normalized to body mass, however, the obese adolescents appear weaker due to reduced mobility, neural adaptations, and changes in muscle morphology [53]. Likewise, reduced power output of EDL, tibialis anterior, and soleus muscles was reported in obese animals despite greater muscle mass gain [54,55]. This trend is not surprising considering the nature of the muscle fibers. Soleus muscles are responsible for postural control and proposed training stimulus evoked by the elevated body mass. This increase in mass does not transfer to an increase in strength, suggesting that the adaptation to greater weight-bearing capacity does not necessarily transfer to an improvement in locomotor performance [54]. Conversely, the maximal isometric stress and normalized power output are significantly reduced in obese EDL muscles, suggesting that larger muscles with decreased function are formed to maintain the equal absolute contractile performance as lean controls [54]. These findings suggest that locomotor function is more likely to be impaired in older obese individuals, with an association between greater body mass and reduced normalized power output from the skeletal muscle [55]. As discussed in the previous section, diet-induced obese individuals have an increased percentage of oxidative muscles resulting from high consumption of fat, which affects the nutrient utilization and oxidative capacity of the oxidative muscle fibers.

The obesity-induced aberration of circulating factors (e.g., insulin, growth hormones, IGF-1, androgens, and pro-inflammatory cytokines) impacts protein synthesis and degradation [56]. This imbalance was shown in human participants where a lower skeletal muscle IGF-1 mRNA and protein were observed in obese men and women [57]. Similarly, obese mice showed reduced IGF-1 mRNA abundance compared to the lean counterpart, implying impaired protein synthetic signaling [58]. Furthermore, this downregulation of IGF-1/Akt signaling, coupled with inflammatory responses, was shown to alter skeletal muscle regeneration in obese mice following injury [58].

Obesity is characterized by systemic inflammation with increased circulating pro-inflammatory cytokines [59]. On the previous point, emerging studies suggest that chronic low-grade inflammation and accumulation of toxic lipid metabolites with obesity can also impair muscle regeneration [60]. Specifically, these toxic lipid metabolites and circulating inflammatory cytokines inhibit IGF-Akt signaling [60] and increase the activation of NFκB signaling and the consequent upregulation of MuRF-1 and MAFbx1 [25]. These alterations can directly impact muscle protein regenerative response while accelerating muscle protein turnover.

Lastly, there is a growing body of evidence suggesting that several secretomes that play a significant role in muscle atrophy are increased in skeletal muscle of extremely obese individuals. For instance, the expression of myostatin, a major regulator of muscle growth, was significantly increased in extreme obesity at the cellular, organ, and systemic levels [61]. A recent cross-sectional study by Kurose and colleagues also reported an elevated serum myostatin level in obese patients [62]. Moreover, this study reported an association of adiponectin with lower skeletal muscle strength, which emphasizes a possible crosstalk between adipose and muscle tissues and the autocrine/paracrine effects of adipokines and myokines that can regulate muscle metabolism [62]. 

### 2.3. Diabetes Mellitus

Nearly half a billion people live with diabetes mellitus (DM) worldwide [63], and in the United States alone, 34.2 million people had DM in 2018 [64]. In fact, DM was the sixth leading cause of disability in 2017 [63] and resulted in a substantial global health economic burden of USD 300 billion. [64]. There is growing evidence suggesting that the cause of the DM epidemic, particularly type 2 DM (T2DM), is linked to the type of diet and lifestyle. In an epidemiological study, the Western dietary pattern was shown to be associated with an elevated risk of developing T2DM [65]. This finding further suggests the negative effect of the Western-style diet pattern on the health of the current population and warrants diet intervention in those with T2DM.

DM is a complex metabolic disorder characterized by hyperglycemia from pancreatic failure and peripheral insulin resistance [63]. Insulin acts as an anabolic hormone and plays an important role in metabolism of carbohydrates, lipids, and proteins. Consequently, long-term and poorly controlled diabetes can negatively affect nutrient delivery and utilization in skeletal muscle, adipose tissue, and the liver. In skeletal muscle alone, absence of insulin can contribute to reduced muscle mass and function through decreased protein synthesis [66]. Diabetic muscular atrophy is a complication that can lead to sarcopenia, high morbidity, mortality, and poor prognosis, and warrants strategies to reduce the health system burden and improve quality of life for patients with DM [67]. There are several classifications for diabetes mellitus suggested by the American Diabetes Association. This review discusses the prevalence of skeletal muscle atrophy in two major classes of DM.

#### 2.3.1. Type 1 Diabetes Mellitus

Type 1 diabetes mellitus (T1DM) is caused by an autoimmune destruction of the pancreatic β-cells leading to absolute insulin deficiency [68]. Although T1DM only accounts for less than 10% of all instances of diabetes mellitus [69], its complications can have a detrimental effect on major organs in the body, including skeletal muscle. In a cross-sectional study, the prevalence of sarcopenia and low handgrip strength in older adults was higher in patients with T1DM than those with T2DM [70]. Emerging studies point to impaired skeletal muscle health and fiber diameter in individuals with T1DM [71], and in hospital settings, severe cases of T1DM are associated with residual weakness after discharge [67]. Moreover, as it is a genetic and autoimmune disorder that accounts for 80–90% of diabetes in children and adolescents [72], muscle atrophy and low bone mineral density are implicated in young adolescents with T1DM [68]. Furthermore, longer duration of the diabetic condition may contribute to the development of osteoporosis in adults [68], which merits early diagnosis and an individualized therapeutic regimen. 

Streptozotocin (STZ) is an alkylating antineoplastic agent that mediates pancreatic islet β-cell destruction and thereby ablation of insulin. By this mechanism, injection of STZ is a widely used approach to produce a model of T2DM or T2DM in rodents. STZ is also used in rodents to examine the consequences of the diabetic condition in skeletal muscle. For instance, 14 days of intraperitoneal injection of STZ in mice resulted in hyperglycemia accompanied by hyperinsulinemia as well as a loss of body mass and skeletal muscle mass [73]. In a study by Ishida and colleagues, mice injected with STZ exhibited hyperglycemia and a significantly reduced gastrocnemius muscle fiber cross-sectional area [74]. In this study, atrophy of muscle was associated with increased serum levels of inflammatory cytokines and ubiquitin ligases in muscle [74], suggesting increased muscle tissue degradation. Similarly, female mice injected with STZ for two weeks also demonstrated weight loss, extreme fat loss, dyslipidemia, ketosis, and skeletal muscle atrophy, which was not prevented by exercise training without insulin therapy [75]. The authors of this study observed an increase in *Akt* gene expression in the muscles, which could be a compensatory response to counteract skeletal muscle atrophy, and downregulation of glucose transporter protein 4 (GLUT4) [75].

The effects of T1DM on protein metabolism seem to be clear as insulin deficiency leads to a profound increase in catabolism [76], reduced motility of myosin, and altered mitochondrial ultrastructure and function [71,77,78]. Recent findings demonstrate mitochondrial dysfunction as a critical mechanism of muscle atrophy in T1DM patients. Specifically, a significant increase in mitochondrial H_2_O_2_ emission and decreases in mitochondrial respiration were observed in a T1DM model [71]. It is notable that T1DM patients with a severe myopathy present alterations in skeletal muscle mitochondrial function and inflammation, which are not ameliorated by physical exercise [78]. These changes are observed in the early phase of the disease, and chronic deterioration of muscle function is expected with long-standing diabetes in adulthood [68]. Thus, an efficient treatment plan is necessary among adolescents to prevent skeletal muscle atrophy and subsequent complications. 

The primary goal of DM treatment is to control glycemia and its complications through lifestyle modification or insulin treatment for patients with T2DM and T1DM, respectively. Controversially, in a cross-sectional study, insulin treatment was one of the risk factors for age-related muscle atrophy in those with diabetes [70], while in a longitudinal study, insulin treatment and correction of glycemic level attenuated the progression of sarcopenia [79]. These conflicting findings suggest that insulin treatment itself may not be the main contributor to accelerated muscle atrophy in older individuals with diabetes. Thus, potential management therapies and future studies should also target other factors such as elevated levels of inflammatory cytokines, mitochondrial dysfunction, and myokine releases to prevent or delay the rate of muscle atrophy in patients with diabetes.

#### 2.3.2. Type 2 Diabetes Mellitus

Individuals with type 2 diabetes mellitus (T2DM) exhibit hyperglycemia as the result of increased hepatic glucose production, peripheral insulin resistance, relative insulin insufficiency and β-cell dysfunction, ultimately leading to β-cell failure [80]. Skeletal muscle, due to its mass, is critical for whole-body insulin-stimulated glucose disposal. Thus, as with T1DM patients, glucose uptake via GLUT4 and metabolism of glucose by muscle are reduced by insulin resistance [81]. Indeed, GLUT4 density in insulin-sensitive slow-twitch muscle fibers of T2DM patients was significantly lower than that of lean counterparts [82]. This suggests that insulin-mediated glucose uptake is partly related to the oxidative capacity of the muscle fiber [83]. Furthermore, muscle insulin receptor knockout mice had significantly smaller fast-twitch glycolytic muscle fibers than those of wild-type mice. Interestingly, though, these mice exhibited decreased relative mRNA expressions of ubiquitin ligases, Atrogin-1, and MuRF1 [66]. These outcomes indicate that insulin resistance-mediated muscle atrophy may be primarily due to decreased protein synthesis rather than proteasomal atrophic pathways. 

Generally, oxidative type I fibers are more sensitive and responsive to insulin as compared to glycolytic fibers in both human and animal subjects [82,83]. Indeed, reduced oxidative enzyme capacity of skeletal muscle is prevalent in patients with either form of diabetes [84], resulting in a reduction of oxidative muscle fibers. In fact, glycolytic capacity is higher in skeletal muscle of patients with T2DM [84], which demonstrates that diabetes-associated muscle atrophy may result from changes in fiber composition and fiber-specific metabolism [84]. 

T2DM is a systemic inflammatory disease. Levels of pro-inflammatory cytokines such as TNF-α and IL-6, which are involved in the development of insulin resistance in skeletal muscle [85], are elevated in T2DM. Increased systemic levels of TNF-α in T2DM increases the inflammatory pathways in muscles, particularly the NFκB pathway, which was reported to contribute to T2DM-mediated muscle atrophy [85]. Furthermore, an increased level of blood cortisol was observed in T2DM patients [86]. Cortisol regulates inflammation, glycogenolysis, and gluconeogenesis, and increased levels of cortisol can lead to hyperglycemia. It also inhibits the actions of insulin and decreases the cellular uptake of glucose in the brain, red blood cells, and skeletal muscle [87]. In a study by Kamba and colleagues, increased GC levels were associated with decreased insulin secretion [88]. Thus, high cortisol levels can accelerate muscle atrophy in diabetic patients. In fact, a physiologic dose of GCs was shown to accelerate muscle protein degradation in mice [89]. 

Further, elevated GC levels have been shown to inhibit protein synthesis and promote protein breakdown [89]. The anti-anabolic effect of GCs is associated with the downregulation of the protein synthesis signaling pathway [90,91]. Specifically, GC binding to its receptor on the glucocorticoid response elements activates the expression of two target genes, REDD1 (regulated in development and DNA damage responses 1) and KLF15 (Kruppel-like factor 15). As proteins are known to repress mTOR signaling, upregulation of REDD1 and KLF15 inhibits mTOR and protein synthesis [92]. Additionally, it has been suggested that GCs may directly interact with PI3K, thereby downregulating the IGF-1 signaling pathway [89] or directly altering the production of muscle IGF-1 [90]. 

The classic actions of GCs on muscle tissue depend not only on plasma levels and regulation via negative feedback mechanisms involving the hypothalamus–pituitary unit, but also on the intracellular activity of 11β-hydroxysteroid dehydrogenase type 1(11β-HSD1) [93]. In skeletal muscle, 11β-HSD1 converts inactive cortisone to active cortisol (corticosterone in rodents), thus regulating the availability of cortisol within the tissue [94]. Increased cortisol availability in liver and adipose tissue from increased expression of 11β-HSD1 is implicated in the development of insulin resistance, obesity, hypertension, and Cushing’s syndromes [95]. Further, overexpression of 11β-HSD1 has been described in muscle weakness and myopathy [96] and selective deletion of this enzyme in mice prevented skeletal myopathy by decreasing local pre-receptor availability of GCs [97]. Muscle weakness, expressed as grip strength, decreased mass of type IIb fibers, and the increased expression of MuRF1, MAFbx, FoxO1, and myostatin were largely prevented in 11β-HSD1 knockout mice treated with corticosterone, suggesting that local GC availability may be involved in the development of muscle myopathy [97]. Selective inhibition of 11β-HSD1 also prevented protein degradation by corticosterone in both murine C2C12 myotubes and primary human myoblasts, and effectively suppressed the effect of corticosterone on myoblast proliferation [98]. Interestingly, recent evidence indicated that increased activity of 11β-HSD1 induced a robust anti-inflammatory effect and protected against skeletal muscle wasting in the TNF-α transgenic murine model of chronic inflammation [99]. The results of these studies indicate that GCs produced locally by 11β-HSD1 can be muscle-sparing in the presence of inflammation and mediate muscle wasting by well-defined protein degradation pathways. 

In a chronic diabetic condition, loss of muscle mass and function are apparent in the appendicular lean muscle mass and trunk [76,100,101] and lower extremities in older adults. Moreover, one study observed a significant reduction of muscle strength in the ankle flexors, ankle extensors, and knee flexors in patients with T2DM, while elbow and wrist muscle strength were preserved [102]. In addition to strength loss, older diabetic adults also exhibit reduced gait speed [103]. This imbalance between the upper and lower extremities in diabetic patients indicates polyneuropathy that presents with motor [104,105] and mitochondrial dysfunction [71,77,106]. 

Diabetic neuropathy is a length-dependent degeneration of nerve fibers that occurs commonly in older diabetic patients. In its most severe form, diabetic neuropathy can result in weakness and atrophy in the lower extremities [102,104]. Interestingly, older adults with T2DM were shown to have an approximately 50% faster rate of decline in leg muscle mass and gait speed compared to those without DM. This suggests that diabetic status may exacerbate the decline in muscle mass and quality already associated with aging [103]. Older adults with diabetes may have greater arm and leg muscle mass than those without diabetes due to their bigger body size; however, older diabetic adults have decreased muscle strength as strength is independent of muscle mass [107]. 

A longer duration of diabetes and poor glycemic control are associated with worse muscle quality [107], suggesting the importance of diabetes management in preventing age-related muscle loss in older adults. Metformin is a commonly prescribed medication for T2DM patients to decrease hepatic glucose production and intestinal absorption of glucose. Intriguingly, a long-term administration of this ‘gold standard’ drug was shown to accelerate the rate of muscle atrophy among T2DM patients by upregulating the expression of myostatin and activation of phosphorylated AMPK [108]. In contrast, Hasan and colleagues reported that metformin-treated mice showed a significant decrease in FoxO3a, MAFbx, and MuRF1 expression, suggesting a beneficial effect of metformin in ameliorating muscle atrophy [109]. This discrepancy may be due to the state of health condition of the study model, one being a diabetic model while another being a diet-induced obese model. Nevertheless, metformin should be prescribed to those with T2DM as it has a profound effect on reducing hyperglycemia and other diabetes-related complications. In addition to metformin, non-pharmacological therapies such as regular exercise and healthy diet should be emphasized to slow the progression of muscle atrophy. 

### 2.4. Sarcopenia

Muscle mass begins to decline by 50 years of age, and becomes more dramatic beyond age 60 [110]. On average, the incidence of muscle loss increases up to 50% in adults over 80 years of age [111], but varies according to sex and physical activity level [112]. Interestingly, older women generally lose more mass than men, but men lose almost twice as much muscle strength as women [113]. The decreased muscle mass and function contribute to the high incidence of accidental falls observed among older adults. Furthermore, functional impairment of muscle mass has consequences for older adults such as inability to participate in physical activities successfully, inability to perform tasks of daily living, and a poor quality of life. Moreover, age-related muscle atrophy has been linked to several chronic conditions such as osteoporosis, insulin resistance, and arthritis [110]. Thus, gaining deeper understanding of the cellular and metabolic etiology of sarcopenia is important.

Sarcopenia, as defined by the European Working Group on Sarcopenia in Older People (EWGSOP), is a syndrome characterized by a progressive loss of skeletal muscle mass and strength with a risk of adverse health outcomes such as physical disability and frailty [114]. It is important to note that sarcopenia means a decline in muscle functionality combined with age-related decreased muscle mass. In fact, loss of muscle strength occurs much more rapidly than the concomitant loss of muscle mass [112], which further suggests a decline in muscle quality with aging [113]. For example, the loss of muscle strength, observed by a grip strength test, starts to decline at 30 years of age, and this rate accelerates after age 40 [115]. In a clinical terminology, sarcopenia is specified as appendicular skeletal muscle mass divided by height in meters of more than two standard deviations below the average of younger persons [116]. 

At a cellular level, the number of muscle fibers and their cross-sectional area decrease during aging. Age-related muscle atrophy is most significant in glycolytic type II muscle fibers than oxidative type I fibers [117]. Thus, the progressive loss of glycolytic type II muscle fiber leads to an increased ratio of type I fibers in older adults [112]. A recent finding by Kim et al. showed that inhibition of lipoxygenases—arachidonate 5-lipoxygenases (Alox5) in particular—prevented the progression of muscle atrophy in aging muscle. In this study, inhibition of Alox5 upregulated the expression of IGF-1, suggesting a new regulator of protein synthesis pathway [118]. Furthermore, a previously characterized liver regeneration drug, malotilate, was shown to preserve fast-twitch fibers in aged muscle by inhibiting Alox5 [118].

Muscle cells repair and regenerate through the action of satellite cells [119]. During physiological or pathological conditions, satellite cells are activated and proliferate myocytes. However, the satellite cell number is reduced in older individuals compared to their younger counterparts [120]. Interestingly, this reduction appears to preferentially affect type II fibers, thus supporting the concept that there is a higher degree of atrophy in glycolytic muscles among older individuals. Recent studies have revealed the role of small extracellular vesicles (sEVs) in the reduction of satellite cells in aging muscles. Specifically, sEVs released by skeletal muscle play an important role in cell–cell communication by delivering biomolecules such as microRNAs (miRNAs) to recipient cells and altering their functions [121]. A study by Shao and colleagues [122] showed that atrophic myotube-derived sEVs *miR-690* may inhibit satellite cell differentiation by targeting myocyte enhancer factor 2 during aging. This finding suggests *miR-690* as a possible therapeutic target for restoring myogenic capacity of satellite cells in aging muscle.

Sarcopenia can occur due to other factors, including an imbalance in protein synthesis caused by inadequate energy intake or change in nutrient metabolism (decreased digestive enzymes). Endocrine dysfunction involving insulin, IGF-1, and cortisol [123] has also been suggested by the literature. For instance, reduction in the production of IGF-1 with aging may directly affect motor neuron integrity, further contributing to aging-associated loss of muscle fiber number and size [124]. Conversely, increased myostatin mRNA expression was observed in older adults compared to younger counterparts, which was associated with reduced muscle area [125]. The authors of this study suggested that myostatin levels in the skeletal muscle may be one of the driving factors for sarcopenia.

Aging is associated with subclinical levels of inflammation that may cause muscle atrophy. For instance, higher circulating pro-inflammatory cytokines such as TNF-α have been reported in older adults [126]. Likewise, the increased level of the pro-inflammatory cytokines appears to affect satellite cell activation and number in older adults [127]. Moreover, IL-6 was shown to play a central role in the hepatic production of C-reactive protein (CRP) [126,128]. Elevated levels of IL-6 and CRP were demonstrated to be associated with 2- to 3-fold greater risk of losing more than 40% of muscle strength in older adults [126]. 

Several cross-sectional studies reported an importance of dietary pattern in aging muscles. In some studies, adherence to an anti-inflammatory Mediterranean diet was inversely associated with sarcopenia as shown by better physical function and reduced self-reported falls [33,129], but neither of these studies found an association between the Western dietary pattern and sarcopenia. This is surprising because the Western diet pattern is generally pro-inflammatory and expected to contribute to the likelihood of developing or worsening sarcopenia. This unexpected finding could be due to other factors such as participants’ weight at baseline, age, and gender. Indeed, in another cross-sectional study, a Western diet was significantly associated with sarcopenia among obese or overweight women [130]. Overall, better diet quality [131] and variety [129], consisting of mostly fruits, vegetables, whole grains, and fatty fish, were shown to be beneficial for muscle quality and function in older adults. 

Other factors contributing to sarcopenia have been discussed, including impaired neuromuscular function (denervation) [124], apoptosis [132,133], cachexia [134], mitochondrial dysfunction [135], and muscle disuse [114,116]. Correspondingly, individuals with pathological conditions such as T2DM have a higher prevalence of sarcopenia than their healthy counterparts [136]. In general, these findings indicate that physical conditions affecting skeletal muscle can accelerate the protein breakdown in aging muscles. 

### 2.5. Alzheimer’s Disease

Alzheimer’s disease (AD) is a slow onset, neurodegenerative disorder that is becoming more prevalent in aging populations worldwide [137]. AD is characterized by progressive memory loss and cognitive impairment and currently, there is no precise test available to diagnose the disease or effective treatment for this condition. Symptoms of cognitive decline, the first clinical manifestation of AD, include reduced interest in the surrounding environment. This results in a dramatic decrease in physical activity and general locomotor activity and can contribute to the development of sarcopenia [138], reduced autonomy, a greater risk of falling, decubitus ulcers, and systemic infection [138]. Evidently, muscle wasting and loss of muscle strength are commonly observed in AD patients measured by magnetic resonance imaging (MRI) and dual emission X-ray absorptiometry (DEXA) and this loss is correlated with hippocampal atrophy and cognitive performance [139]. 

According to a community-based cohort study, individuals who develop AD are older and have a lower cognitive function and decreased strength in several muscles compared to those with normal cognitive function [140]. In this study, greater muscle strength was associated with a decreased risk of developing AD. Similarly, a cross-sectional study by Low et al. linked lower limb muscle mass with immediate memory, delayed memory, and constructional ability among older adults (mean age 61.4 years) with T2DM [141]. Another study reported an association of decreased muscle strength, but not muscle mass, with a decline in cognition [142]. Moreover, physical difficulties, such as decreased hand grip strength and low gait speed, were observed in elderly individuals with cognitive impairment and dementia. These patients also experienced reductions in muscle strength of both upper and lower extremities [143]. Overall, these findings suggest that skeletal muscle mass may be a useful marker of possible co-occurring cognitive dysfunction and that skeletal muscle mass may play a protective role against cognitive diseases.

An overview of pathological mechanisms explaining the association of muscle loss and cognitive impairment is described thoroughly by Oudbier et al. [144]. Briefly, myokines, such as fibronectins, are secreted by physically active muscle and upregulate neurotrophic factor in the brain microenvironment. Dysfunctional myokine secretion due to physical inactivity results in systemic inflammation and muscle glucose metabolism, potentially affecting the transport of insulin across the blood–brain barrier. Furthermore, dysfunctional mitochondria in skeletal muscle can result in excessive production of reactive oxygen species, worsening oxidative stress, and lead to a decline in cognition [144]. 

On the other hand, the etiology of AD development is still unclear. Currently, the most accepted pathological hallmarks of AD include amyloid-beta (Aβ) accumulation and the presence of phosphorylated tau (τ) in the brain [140]. Emerging clinical studies utilize technologies with superior sensitivity and specificity to measure blood-based Aβ biomarkers and phosphorylated τ [145]. Using this technology, a large body of studies report a likelihood of developing mild cognitive impairment or dementia in cognitively normal participants with Aβ biomarkers than those without Aβ biomarkers [146], making Aβ biomarkers an important predictor of AD dementia prognosis. Interestingly, these features are not restricted to the brain and can be seen in skeletal muscle [147]. 

Recent animal studies have linked the accumulation of Aβ and its precursor within muscle fibers with increased muscle atrophy. In a study by Lin et al., a double transgenic amyloid precursor protein and presenilin 1 (APP/PS1) mouse model of AD exhibited lower body weight and lean tissue mass than the sex- and age-matched counterparts [148]. Additionally, muscle atrophy and the extent of memory decline were strongly associated in the APP/PS1 mice [148,149]. The authors described this phenomenon as the mitochondrial respiratory deficits that occur in the brain and muscle tissue of the AD-relevant transgenic mice [149]. In fact, a significant decrease in the mitochondrial function in both soleus (oxidative) and plantaris (glycolytic) muscles was observed in these transgenic mice [149]. Similarly, age-dependent accumulation of Aβ peptide occurred in the skeletal muscle of triple transgenic (3xTg)-AD mice that preceded bioenergetic mitochondrial dysfunction. However, this was only detected at 12 months of age, and characterized by a decreased respiratory control ratio and impaired complex I activity [135]. In a similar vein, an investigation using muscle biopsies of patients with myositis showed an upregulation inflammatory cytokine IL-1β, which led to the accumulation of Aβ [150]. Indeed, mitochondrial functions and inflammatory signals are directly linked to AD symptoms and pathogenesis, which is explained in a review by Yoo et al. [151]. Thus, future studies should further investigate the contribution of skeletal muscle mitochondria and inflammatory signaling in the development of AD and address whether reducing mitochondrial dysfunction-mediated inflammation could attenuate the progression of AD.

Exercise directly impacts the brain health. Although the exact mechanisms have not been defined, past findings support the direct crosstalk between muscle and brain, mediated by exercise-stimulated release of muscle-derived molecules [152]. Skeletal muscles are important secretory organs, communicating toxic and metabolic stress to the central nervous system via secretion of bioactive molecules collectively known as myokines [153]. Myokines can both be pro- and anti-inflammatory and include cytokines, peptides, and growth factors including myostatin, IL-6, IL-15, myonectin, fibroblast growth factor (FGF) 21, and brain-derived neurotrophic factor (BDNF), which are described in detail elsewhere [154]. Myokines are synthesized and released by myocytes in muscle tissue in response to muscular contractions [154]. Some myokines, such as IGF-1, are secreted by muscle during exercise and are also increased in the brain, which induces beneficial effects [155]. Atrophying muscle, on the other hand, suppresses the endocrine function of myocytes and favors the release of unbeneficial inflammation, worsening dementia and cognitive decline [156]. Importantly, past studies show that skeletal muscle atrophy induced by lack of use initiates the onset of memory dysfunction even prior to the deposition of Aβ [157]. 

Although results remain controversial, emerging studies have associated exercise with Aβ clearance and reduced cognitive decline, possibly by modulating microglia-mediated neuroinflammation and oxidative stress [158]. Likewise, exercise can beneficially improve cellular markers associated with AD, thereby reducing the accumulation of Aβ plaques [159]. Lee et al. reviewed how regular exercise is associated with the release of myokines that exert beneficial effects on neurodegenerative diseases through various mechanisms including cell survival, neurogenesis, neuroinflammation, proteostasis, oxidative stress, and protein modification [160]. Together, regular exercise seems to have a protective effect against the development of AD or other neurodegenerative diseases by inhibiting different pathophysiological mechanisms. 

Understanding the interrelation between skeletal muscle and AD development is crucial as neurodegenerative symptoms increase the burden of the disease and worsen the quality of life for the patients and their caregivers [148]. Overall, current studies indicate that AD pathologies with Aβ accumulation are not exclusive to brain tissue and that skeletal muscle dysfunction and loss tend to occur prior to the exhibition of clinical symptoms of AD [135,139]. These findings suggest muscle mass as a potential clinical diagnostic measure for early detection of AD and signify the utmost importance of exercise and preservation of skeletal muscle mass in the older age population. 

### 2.6. Cancer Cachexia

Cachexia is a multifactorial syndrome characterized by a continuous skeletal muscle loss with or without loss of adipose tissue that cannot be corrected with nutritional support [161]. Cachexia generally manifests in chronic diseases such as cancer, chronic obstructive pulmonary disease (COPD), chronic heart failure (CHF), and chronic kidney disease (CKD) [162]. About 1.3 million people in the U.S. with cancer are affected by cachexia [134]. Cachexia and associated skeletal muscle loss may be present earlier in cancer progression [126], demonstrating reduced muscle fiber size and protein content [163]. This loss, especially of visceral protein and lean body mass loss due to an increased rate of protein turnover, is a prognostic factor in determining the overall survival of the patients [164] because reduction in muscle protein may culminate in conditions of asthenia, immobility, and cardiac or respiratory failure [165]. Furthermore, cachexia reduces the effectiveness of cancer treatment [166].

The more apparent reason for a significant weight loss in cachectic patients may be inadequate food intake or anorexia. However, reducing food intake alone may not be the sole cause of muscle loss, as nutritional supplementation did not prevent weight loss in cachectic patients [166]. From a metabolic perspective, the release of amino acids from muscles becomes a beneficial substrate for gluconeogenesis and acute-phase protein synthesis. This provides an important energy source for energy enterocytes and the immune cells [167]. However, due to the nature of cancer cells being highly proliferating and energy-demanding [168], cancer patients are in a negative energy balance, further encouraging tumor growth [169]. Furthermore, there is an increased energy use in cachectic patients associated with futile cycles in the triacylglycerol/fatty acid substrate cycle and the Cori cycle, and increased expression and activity of mitochondrial uncoupling proteins [170].

Although the exact molecular mechanisms of cancer cachexia are unclear, several key mediators, including pro-inflammatory cytokines, have been shown to be elevated in cancer patients. These cytokines include the TNF-α, IL-1β, IL-6, and interferon (IFN)-γ [134,163,165,171], which mediate muscle atrophy via the ubiquitin-dependent proteasome (UPS) pathway [167]. In recent findings, cancer-derived extracellular vesicles (EVs) have been found to play a role in the development of cancer cachexia. For instance, cancer-derived EVs can target cells and transfer their contents, thereby regulating induction of cancer cachexia [172]. In a recent study, inhibiting EV secretion attenuated muscle wasting and adipose degradation in cachexic model [172]. Myostatin can also play a major role in a cachectic state. In a mouse study, a myostatin inhibitor mitigated the weight loss induced by cancer cachexia and protected the muscle cell morphology. Furthermore, the myostatin inhibitor decreased the expression of ubiquitin ligases in the quadriceps in vivo [173]. These findings suggest involvement of various molecular factors in cancer cachexia-induced muscle atrophy. 

### 2.7. Heart Failure

Heart failure (HF) is highly prevalent in older adults and its incidence continues to rise over time. According to 2015–2018 data, about 6 million American adults over 20 years of age had HF [174]. HF may result in cardiac cachexia, implying that HF is associated with a significant weight loss syndrome [175]. Combined with a low peak oxygen consumption, patients with cardiac cachexia are at an extremely high risk of death [176]. One study described that muscle wasting among cardiac patients was mainly due to decreased caloric and protein intake, suggesting that diet history may be a simple technique that may be an essential measure [177].

Individuals with chronic HF exhibit skeletal muscle abnormalities and early fatigue, involving the large locomotive muscles. These abnormalities can be seen in handgrips and quadriceps [178,179]. Patients exhibit reduced skeletal muscle volume, strength, and endurance [180]. Specifically, decreased oxidative capacity of working muscles [177,181] and a shift from predominantly fatigue-resistant oxidative towards glycolytic muscle fibers [180] are observed in HF patients. The decrease in the oxidative capacity may be explained by disuse of muscles or mitochondrial dysfunction combined with abnormal energy metabolism [182], further contributing to early lactic acidosis and fatigue with exercise [180,183]. In fact, only the mitochondrial enzymes involved in terminal oxidation are decreased in the HF patients. In contrast, the total phosphorylated and glycolytic enzyme activities were not affected, which explains the phenomenon of a decreased percentage of the slow-twitch type I fibers in HF patients [183].

TNF-α, myostatin, and hormones such as epinephrine, norepinephrine, aldosterone, and cortisol, are upregulated in HF and could contribute to HF-associated myopathy development [184]. Although the initiating factors of muscle inflammation are not completely understood, disuse-induced atrophy may promote inflammation by activating several signaling pathways, including the NFκB signaling pathway [182]. Several studies have indicated the role of myostatin in muscle atrophy in HF. Myostatin may be released directly from cardiomyocytes in HF [184]. Indeed, upregulation of plasma myostatin is reported in patients with HF, and inhibiting myostatin in cardiac cachectic mice promoted greater maintenance of muscle mass [184]. Additionally, HF may be associated with insulin resistance independent of HF etiology [185]. Other specific mechanisms of muscle wasting in HF individuals remain to be elucidated. 

## 3. Concluding Remarks

Skeletal muscle is the primary organ responsible for not only physical activities, but also for nutrient metabolism, whole-body glucose homeostasis, and insulin sensitivity. Thus, loss of muscle mass and function can damage overall health. Muscle atrophy is implicated in various pathophysiological conditions including obesity, diabetes, sarcopenia, Alzheimer’s disease, cancer cachexia, and heart failure (Table 1). A Western diet pattern, also known as a standard American diet, is shown to accelerate the rate of muscle atrophy. Various mechanisms and stimuli contributing to muscle atrophy have been discussed in the literature. Notably, oxidative stress, mitochondrial dysfunction, inflammatory cytokines (TNF-α, IL-1β, IL-6), glucocorticoids, and myostatin have been implicated in atrophying skeletal muscle, making these molecular factors considerable therapeutic targets. Currently, most studies on muscle degeneration are being done in animal subjects and in vivo. Future studies should include more human participants to determine how different types of muscle fibers are affected by these various stimuli. This will provide better therapeutic treatment options that target a specific muscle fiber type in different physiological conditions. This review article discussed several mechanisms that lead to muscle atrophy (Figure 1); however, there may be other contributors that are not included in this review. Future studies are needed to investigate potential therapeutic agents to prevent or delay skeletal muscle atrophy in those with chronic conditions. 

## Figures and Tables

**Figure 1 ijms-24-02973-f001:**
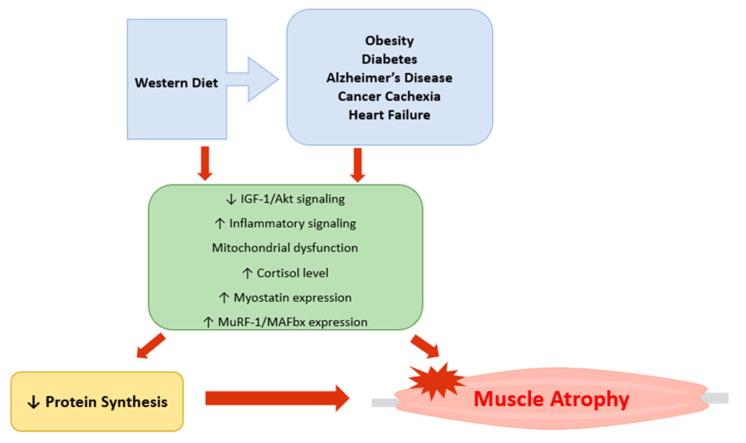
Graphic representation of the major mechanisms involved in skeletal muscle atrophy. Western diet pattern, obesity, diabetes, Alzheimer’s disease, cancer cachexia, and heart failure reduce protein synthesis by inhibiting IGF-1/Akt signaling and increasing inflammatory signaling, myostatin expression, and atrogenes (MuRF-1 and MAFbx). Mitochondrial dysfunction has also been implicated in development of muscle atrophy. Imbalance between the rate of protein synthesis and degradation can also lead to muscle atrophy.

**Table 1 ijms-24-02973-t001:** Prevalence of skeletal muscle atrophy.

Condition	Affected Muscle	Main Findings	References
*High-Fat Diet*	EDL muscle	More than 12-week HFD induced degradation of fast-twitch muscle fibers (EDL).	[16,20]
	Gastrocnemius muscle	Long-term HFD decreased mitochondrial enzyme gene expression and activity in the gastrocnemius muscle.	[19]
	Soleus muscle	Three-week HFD induced denervation muscle atrophy in soleus muscle.	[17]
	EDL and soleus muscles	Shift from type IIb fiber to type IIa fiber was observed in the *ob/ob* group in the EDL muscles, whereas the opposite was observed in the soleus muscles.	[186]
*Obesity*	Soleus and EDL muscles	The absolute isometric force of the obese soleus muscles was significantly greater than that of lean controls; however, the maximal isometric stress and normalized power output of both obese soleus and EDL muscles were reduced.	[54]
	Soleus and EDL muscles	Despite the increased muscle mass in the HFD-induced obese soleus and EDL muscles, muscle strength remained unchanged.	[55]
	EDL, gastrocnemius, and plantaris muscles	EDL, gastrocnemius, and plantaris muscles were significantly smaller in the obese animals compared to lean counterparts.	[47]
	EDL and soleus muscles	EDL and soleus muscle mass were lower in the obese Zucker rats than lean controls, concomitant with reduced fiber area.	[187]
*Diabetes Mellitus*	Leg and appendicular muscles	T2DM was associated with accelerated loss of leg muscle strength and quality in older adults.	[100,188]
	Appendicular, trunk, and thigh muscles	Older adults with either diagnosed or undiagnosed T2DM showed excessive loss of muscle mass.	[76]
	Total and appendicular muscles	There was a greater loss in total and appendicular lean muscle mass in older men with untreated DM compared to normoglycemic counterparts.	[189]
	Gastrocnemius	Mice with T1DM phenotype showed a significantly lower gastrocnemius muscle fiber cross-sectional area.	[74]
	Ankle flexors, ankle extensors, knee flexors, elbow, and wrist	Significant reduction was observed in the muscle strength of the ankle flexors (17%), ankle extensors (14%), and knee flexors (14%) in patients with T2DM. Elbow and wrist muscle strengths were preserved.	[102]
	Quadriceps	DM status was significantly associated with reduced gait speed. Insulin-dependent older adults had significantly reduced quadricep strength and power.	[103]
	Hand grip and knee extensor	Older adults with T2DM had lower muscle strength, but not muscle mass compared with non-diabetic counterparts.	[107]
*Sarcopenia*	Gastrocnemius, EDL, and soleus muscles	Muscle atrophy was greatest in aging gastrocnemius muscles and intermediate in aging EDL and soleus muscles.	[117]
	Leg circumference	The leg circumference of >40 years of age was less than that of <40 years of age.	[190]
	Abdomen, lower extremities	Muscle thickness in the abdomen and lower extremities decreased significantly in Japanese men aged 60 years and older.	[191]
	Hand grip and upper and lower extremities	Decreased hand grip strength and low gait speed were observed in elderly individuals with cognitive impairment and dementia. Reductions in muscle strength of both upper and lower extremities were higher in the AD group.	[143]
	Tibialis anterior muscle	There was a reduction in axonal innervation of transgenic mice compared to the wild-type control.	[138]
*Cancer* *Cachexia*	Soleus and gastrocnemius muscles	Muscles of MAC16 injected mice showed an increased lysosomal protease activity.	[164]
	Quadriceps	Injection of IL-6 in mice with colon cancer induced systemic muscle wasting.	[171]
	Gastrocnemius muscle	There was a significant nitrogen loss with a depressed muscle protein synthesis and increased protein degradation during cancer cachexia.	[192]
	Rectus muscle	Mean muscle fiber diameter in cachectic cancer patients was reduced by about 15% compared to non-cachectic cancer patients.	[163]
*Heart* *Failure*	Type I fibers	Patients with chronic HF developed significant abnormalities in skeletal muscle, reflecting a decreased oxidative capacity in type I fibers.	[181]
	Handgrip and quadriceps	Patients with chronic HF presented reduced muscle mass and decreased exercise capacity in treadmill performance and walking exercise tests.	[179]
	Quadriceps	Myofibril contractile function was strongly related to insulin sensitivity in HF patients, independent of muscle size.	[185]

Abbreviations: T2DM, type 2 diabetes mellitus; DM, diabetes mellitus; STZ, streptozotocin; T1DM, type 1 diabetes; EDL, extensor digitorum longus; HFD, high-fat diet; *ob/ob*, obese mouse; AD, Alzheimer’s disease; IL-6, interleukin-6; HF, heart failure.

## Data Availability

Not applicable.

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
