# Peer review of "Prevalence and Mechanisms of Skeletal Muscle Atrophy in Metabolic Conditions"

_ijms, 2023, doi:10.3390/ijms24032973_

Round 1

Reviewer 1 Report

The review 'Prevalence and Mechanisms of Skeletal Muscle Atrophy in Metabolic Conditions' by Jun, et al, discusses different types of disease and aging-related muscle wasting, in addition to covering the effects of diet, exercise, and metabolic disorders. I believe that this is an extensive review addressing a broad range of topics and is suitable for publication in the International Journal of Molecular Sciences after the following minor revisions:

1) Some recent papers may be mentioned in the review to keep it up to date, such as the following:

For the sarcopenia section (role of micro RNAs and new regulators of the IGF-1 metabolic pathway):

a) Atrophic skeletal muscle fibre-derived small extracellular vesicle miR-690 inhibits satellite cell differentiation during ageing.

Shao X, Gong W, Wang Q, Wang P, Shi T, Mahmut A, Qin J, Yao Y, Yan W, Chen D, Chen X, Jiang Q, Guo B.

J Cachexia Sarcopenia Muscle. 2022 Oct 13.

PMID: 36237168

b) Inhibiting 5-lipoxygenase prevents skeletal muscle atrophy by targeting organogenesis signalling and insulin-like growth factor-1.

Kim HJ, Kim SW, Lee SH, Jung DW, Williams DR.

J Cachexia Sarcopenia Muscle. 2022 Oct 11.

PMID: 36221153

For the cancer cachexia section (role of tumor-derived vesicles in muscle wasting):

a) Atractylenolide I ameliorates cancer cachexia through inhibiting biogenesis of IL-6 and tumour-derived extracellular vesicles.

Fan M, Gu X, Zhang W, Shen Q, Zhang R, Fang Q, Wang Y, Guo X, Zhang X, Liu X.

J Cachexia Sarcopenia Muscle. 2022 Sep 9.

PMID: 36085573

2) Line 48: Minor English correction: put ‘the’ before ‘Western-style’

Line 162: Minor English correction: put ‘a’ before ‘Western style’

Both ‘Western-style’ and ‘Western style’ are used, please choose one form to be consistent.

Author Response

Reviewer # 1

The review 'Prevalence and Mechanisms of Skeletal Muscle Atrophy in Metabolic Conditions' by Jun, et al, discusses different types of disease and aging-related muscle wasting, in addition to covering the effects of diet, exercise, and metabolic disorders. I believe that this is an extensive review addressing a broad range of topics and is suitable for publication in the International Journal of Molecular Sciences after the following minor revisions:

1) Some recent papers may be mentioned in the review to keep it up to date, such as the following:

For the sarcopenia section (role of micro RNAs and new regulators of the IGF-1 metabolic pathway):

  1. a) Atrophic skeletal muscle fibre-derived small extracellular vesicle miR-690 inhibits satellite cell differentiation during ageing.

Shao X, Gong W, Wang Q, Wang P, Shi T, Mahmut A, Qin J, Yao Y, Yan W, Chen D, Chen X, Jiang Q, Guo B.

J Cachexia Sarcopenia Muscle. 2022 Oct 13.

PMID: 36237168

Response: As per the reviewer's suggestion, the reference has been included in lines # 463-471.

  1. b) Inhibiting 5-lipoxygenase prevents skeletal muscle atrophy by targeting organogenesis signalling and insulin-like growth factor-1.

Kim HJ, Kim SW, Lee SH, Jung DW, Williams DR.

J Cachexia Sarcopenia Muscle. 2022 Oct 11.

PMID: 36221153

 Response:  This reference has been included in lines # 452-458.

For the cancer cachexia section (role of tumor-derived vesicles in muscle wasting):

  1. a) Atractylenolide I ameliorates cancer cachexia through inhibiting biogenesis of IL-6 and tumour-derived extracellular vesicles.

Fan M, Gu X, Zhang W, Shen Q, Zhang R, Fang Q, Wang Y, Guo X, Zhang X, Liu X.

J Cachexia Sarcopenia Muscle. 2022 Sep 9.

PMID: 36085573

 Response: This reference is now added in lines # 633-637.

2) Line 48: Minor English correction: put ‘the’ before ‘Western-style’

Line 162: Minor English correction: put ‘a’ before ‘Western style’

Both ‘Western-style’ and ‘Western style’ are used, please choose one form to be consistent.

Response: We thank the reviewer, and all these errors are being corrected.

Reviewer 2 Report

This review “Prevalence and Mechanisms of Skeletal Muscle Atrophy in Metabolic Conditions” from Jun and colleagues deals with the skeletal muscle atrophy and several possible mechanisms. In particular, it emphasises muscle atrophy associated with certain metabolic condition such as high-fat, high-sugar diet patterns, obesity, and diabetes. Other conditions like sarcopenia, Alzheimer’s disease, cancer cachexia have been also considered. Skeletal muscle atrophy is undoubtedly a topic of great interest with a high significance in the health of the modern society that is becoming progressively older.

Authors presented a variety of clinical situations accompanied by atrophy with the proposed underpinning mechanism. A large amount of studies  have been taken into account  and cited (189), and about 25% of them are quite recent (last 5 years).

The table helps to summarize and highlight the information collected.

I feel to recommend the publication of this review in the present form.

Author Response

We thank the reviewer for recommending this article for publication.